# Comparative Analysis of Pigeonpea Stalk Biochar Characteristics and Energy Use under Different Biochar Production Methods

**Nallagatla Vinod Kumar [1], Gajanan L. Sawargaonkar [2,*], C. Sudha Rani [3], Ajay Singh [2], T. Ram Prakash [4], S. Triveni [5], Prasad J. Kamdi [2], Rajesh Pasumarthi [2], Rayapati Karthik [1] and Bathula Venkatesh [1]**

[1] Department of Agronomy, College of Agriculture, Professor Jayashankar Telangana State Agricultural University PJTSAU, Rajendranagar, Hyderabad 500030, India; vinodnallagatla@gmail.com (N.V.K.); karthikrayapati48@gmail.com (R.K.); venkateshbathulavenkychinna@gmail.com (B.V.)

[2] International Crops Research Institute for the Semi-Arid Tropics, Patancheru, Hyderabad 502324, India; ajay.singh@icrisat.org (A.S.); prasad.kamdi@icrisat.org (P.J.K.); rajesh.pasumarthi@icrisat.org (R.P.)

[3] Agricultural Research Station, Vikarabad, Tanduru 501141, India

[4] AICRP on Weed Control, Professor Jayashankar Telangana State Agricultural University PJTSAU, Rajendranagar, Hyderabad 500030, India; trp.soil@gmail.com

[5] Department of Agricultural Microbiology and Bio-Energy, College of Agriculture, Professor Jayashankar Telangana State Agricultural University PJTSAU, Rajendranagar, Hyderabad 500030, India; triveni_agmsc@yahoo.co.in

\* Correspondence: gajanan.sawargaonkar@icrisat.org; Tel.: +91-8455-683-438

**Abstract:** The disposal of crop residues from agricultural fields is often seen as a burden due to the difficulties involved. However, this study aims to turn pigeonpea stalks into biochar, which can serve as a fuel substitute and soil amendment to sequester carbon. Different pyrolysis methods were employed to investigate the variations in yield, physicochemical characteristics, and higher heating value (HHV) of biochar produced from pigeonpea stalks. The biochar produced using a muffle furnace exhibited higher fixed carbon and ash content. These characteristics make it beneficial for restoring degraded agricultural soils by enhancing carbon sequestration. In addition, the muffle furnace biochar demonstrated a total potential carbon ranging from 262.8 to 264.3 g of carbon per kilogram of biochar, along with a $CO_2$ reduction potential ranging from 77.17 to 79.68 $CO_2$ eq per kg. Both the European Biochar Certificate and the International Biochar Initiative confirmed the agronomic abilities of the biochar and its compliance with the highest quality standards for soil carbon sequestration, with 0.11 H/C and 0.7 O/C ratios. Furthermore, biochar produced by muffle furnace from pigeonpea stalks exhibited superior fixed carbon recovery efficiency (181.66 to 184.62%), densification (5.86 to 6.83%), energy density (1.77 to 2.06%), energy retention efficiency (54.80 to 56.64%), fuel ratio (18.95 to 22.38%), and HHV (30.66 to 32.56 MJ kg$^{-1}$). Additionally, it had lower H/C and O/C ratios, suggesting its potential as an alternative solid fuel. The results of the characterization of biochar with scanning electron microscopy (SEM) and Fourier transform infrared spectroscopy (FTIR) revealed that the biochar samples prepared with both the methods had carbonyl (C=O), C=C, and aromatic C-H functional groups; however, the biochar prepared in the muffle furnace had more porosity. In summary, this study highlights the potential of using pigeonpea stalks to produce biochar, which can be utilized as a renewable fuel substitute and soil amendment to sequester carbon. The biochar derived from the muffle furnace exhibited desirable physicochemical characteristics, high carbon content, and excellent energy properties, making it a promising option for various applications.

**Keywords:** muffle furnace; pigeonpea; biochar; kiln; carbon sequestration potential; FTIR; SEM

## 1. Introduction

India is one of the top nations in the world for crop production and is well-known for its vast agricultural landscape and flourishing rural economy. Benefitting from diverse climatic zones and a robust farming sector, India's agriculture significantly contributes to the nation's economy and ensures food security. However, the intensive cultivation methods employed in this agriculturally driven country lead to significant generation of crop residues [1]. According to information from the Ministry of Renewable Energy [2], India generates more than 500 million tons of crop by-products each year. During the period 2017–2018, out of the 516 million tons produced, approximately 116 million tons were incinerated [1]. Effective management of crop residues poses a pressing challenge, necessitating comprehensive assessment and the implementation of strategies to mitigate environmental, economic, and health implications. An estimate claims that, if biomass burning were to be stopped in North India, there might be an economic gain of more than $1.7 billion (in PPP values) over the course of five years due to a decline in the prevalence of hypertension [3]. While some crop residues find utilization in industrial processes, animal feed, roofing material for rural homes, and cooking fuel, a considerable portion remains unutilized and is burned in agricultural fields [4]. Out of the total crop waste of 140 million tons, approximately 92 million tons are openly burned, exacerbating the situation [2]. The prevalence of biomass burning in certain regions of India can be attributed to various factors such as traditional farming practices, limited time windows [5], lack of alternatives, and the economic feasibility of burning these crop wastes to clear the land. Moreover, inadequate awareness and education further contribute to the persistence of this practice, resulting in significant environmental and health impacts. As reported by [6], the burning of 63 Mt agricultural crop stubble, 1.2 Mt of particulate matter (PM), 91 Mt of $CO_2$, 0.6 Mt of $CH_4$, 3.4 Mt of CO, and 0.1 Mt of $NO_x$ into the atmosphere [7]. The open-burning process dramatically increases soil temperatures from 33.8 °C to 44.2 °C, reaching depths of up to 1 cm, which significantly reduces the organic matter in the soil, some of the beneficial microorganisms, and C and N contents [8,9]. This scenario profoundly impacts soil health.

To address these challenges and promote sustainable agricultural practices, the conversion of biomass into biochar has emerged as a promising and environmentally friendly approach. Biochar is an exquisitely textured substance that is akin to charcoal, generated by subjecting diverse biomass and biodegradable waste to pyrolysis. It possesses abundant organic carbon and remarkable resilience against degradation. In terms of surface area, porosity, catalytic activity, and physicochemical stability, biochar outperforms raw biomass materials [10]. The possible advantages of applying biochar have been outlined in several studies, with an emphasis on the agricultural industry [11–14]. It is reported that adding biochar as a soil amendment raises crop yields, reduces fertilizer needs, and boosts soil nutrient density and water-holding capacity. As possible adsorbents that function as a shield and improve the health of the ecosystem and soil in cultivable areas, biomass-based biochar has gained popularity [15]. Carbonaceous biomass is thermochemically transformed to biochar during pyrolysis, which occurs at high temperatures (300–900 °C) and with little oxygen [16]. A comparative study was conducted on biochar produced from different biomasses [17]. The organic parts of paddy straw, bagasse, and wood stem resulted in yields between 24 and 28 weight percent, while cocopeat produced a yield of 46 weight percent. Compared to biochar from rice straw, the biochar derived from rice husk exhibited greater levels of volatile matter and fixed carbon [18].

At present, there are several techniques for biochar production that come under traditional and modern approaches. In traditional approaches, slow and fast pyrolysis techniques are covered, whereas gasification, torrefaction, and hydrothermal carbonization are considered as modern approaches. In slow pyrolysis, biomass is heated to temperatures between 300 and 600 °C at a pace of 5 to 7 °C per minute [19]. Traditional pyrolysis also produces byproducts like syngas and bio-oil in addition to its main product, biochar, which constitutes 35 to 45% [20,21]. Quick pyrolysis offers the benefits of reduced retention time and improved product yield [22]. The rate of heating is greater than 300 °C $min^{-1}$,

and the temperature is above 500 °C in the absence of oxygen during fast pyrolysis [23]. Biochar is also produced by hydrothermal carbonization (HTC), a process that uses high-moisture feedstocks such as animal waste, compost, and sewage sludge [24]. Another method of biochar and syngas production using solid fuel is gasification. Gasification provides lower emissions and larger syngas volume as compared to other conventional methods such as fermentation, pyrolysis, and combustion. In gasification, a major product is hydrogen. However, biochar is also generated in considerable amounts during the gasification process [24]. The production of biochar using a kiln can easily be adopted by smallholder farmers.

Biochar, derived from biomass via pyrolysis, is a carbon-dense substance. While it shows promise as an economical and green solution, its economic and environmental efficacy is influenced by various elements. These include the type of raw material, the production technique, and the method of application. Biochar offers several environmental advantages, including carbon capture, better soil health, minimized nutrient runoff, and decreased pest and disease issues. On the other hand, it poses potential environmental threats, including the release of noxious emissions, contamination of water sources, and contribution to deforestation. Comprehensive research is essential to thoroughly assess the pros and cons of biochar [25]. The profitability and acceptability of biochar production and use depend on a variety of case-specific factors, including location, feedstock, scale, pyrolysis conditions, biochar price, cultivated crop, and potential internalization of externalities, which discourages private investment. Those aspects need to be properly taken into account in each situation to promote biochar development and implementation [26].

Millions of smallholder resource-poor farmers residing in the semi-arid tropics of Asia and Africa depend on pigeonpea cultivation for their food security. It accounts for about 15% of the total pulse harvest from India, and India contributes 70% of total pigeonpea production globally [17,27]. The primary waste generated by this crop is the woody pigeonpea stalk (PPS), estimated to be approximately 2.9 t ha$^{-1}$ or 10.33 million tons annually [28]. The potential of pigeonpea stalks (PPS) as an alternate energy substitute and an organic amendment to soil is underestimated, and most of the PPS produced in India is used as a source of cooking fuel in rural areas [27]. However, PPS holds the potential to be used in the production of biochar and activated biochar, offering an alternative to open burning and inefficient utilization practices that contribute to GHG emissions. Despite limited studies on biochar generation and characterization from PPS, the present study seeks to evaluate different methods for preparing biochar and quantify the characteristics of biochar produced through various techniques.

Previous research has emphasized the influential role of feedstock and production conditions in determining various aspects of biochar, including its yield [29,30], stability [2,29–37], energy properties [38–43], total potential carbon (TPC), physicochemical properties [44–56], and $CO_2$ reduction potential [31,57,58]. Therefore, gaining a comprehensive understanding of the entire biochar production process and conducting thorough characterization is crucial for identifying the most suitable applications for biochar. For instance, biochar with high recalcitrance can serve as an effective carbon sequestration material and, as it possesses high porosity, can be employed as a soil fertility amendment [7]. In order to meet industrial needs, biochar with a higher heating value (HHV) can also be used in the production of solid fuel in the form of briquettes.

To better understand the behavior of pyrolysis and explore new options for producing biochar, a detailed examination of the properties of biomass material is necessary. However, there is currently a shortage of research on the generation and characterization of biochar from PPS. This study aimed to address this gap by evaluating various methods of biochar preparation and measuring the characteristics of biochar produced through different techniques.

## 2. Materials and Methods

### 2.1. Raw Material Collection for Biochar Production

The PPS was collected from ICRISAT (International Crops Research Institute for the Semi-Arid Tropics), situated in Patancheru, Hyderabad, Telangana, at an altitude of 461 m above mean sea level (MSL), at 17°15′ N latitude and 77°35′ E longitude. The raw material of pigeonpea stalks was collected, sliced into pieces (2–5 cm) using a shredder CUM chipper machine (model no. CS50 from Bhide and Sons), dried in an oven (70 °C) for up to 48 h, and screened with 20 mesh sieves for muffle furnace biochar preparation. The biochar kiln unit and its operations pertaining to biochar production (kiln biochar method) were elucidated in detail by [59,60].

#### 2.1.1. Biochar Prepared in the Muffle Furnace (BPMF)

The process involved a muffle furnace in facilitating the production of biochar with higher fixed carbon ash content by heating to an elevated temperature of 500 °C in controlled conditions.

The muffle furnace, equipped with a digital temperature regulator from a Thermo Scientific Thermolyne furnace large tabletop muffle/Atmosphere-Controlled Ashing Model type F30400, was employed to carry out the slow pyrolysis of dried PPS. With a power supply of 220/230 V AC and PID control, the furnace facilitated precise temperature control. A total of 500 g of PPS was subjected to pyrolysis at a consistent temperature of 500 °C while being continuously purged with high-purity nitrogen (99.9%). The temperature was increased at a rate of 10 °C per minute over a duration of 1 h to maintain uniformity throughout the pyrolysis process. Before initiating the pyrolysis, the biomass was placed in the muffle furnace, and, continuously for 10 min, nitrogen gas was purged to ensure an oxygen-deprived atmosphere [61]. The equation used to calculate the yield of biochar (%) or carbon recovery (%) is as follows:

$$\text{Biochar yield (\%)} = \frac{\text{Mass of biochar obtained}}{\text{Mass of feedstock loaded}} \times 100 \tag{1}$$

#### 2.1.2. Biochar Prepared in a Pyrolysis Kiln (BPPK)

The kiln system was employed as another method for biochar production. In this method, the pigeonpea stalks were loaded into a metal container or barrel and heated, initiating the pyrolysis process. The drum system offered a cost-effective and relatively simple approach to biochar production, suitable for small-scale or decentralized settings. However, the drum biochar might exhibit variations in its characteristics due to less precise temperature control compared to the muffle furnace.

In this study, we evaluate how various pyrolysis techniques impact the attributes and potential uses of the resulting biochar. The investigation contrasts the traits and performance of biochar derived from muffle furnaces with those from drum systems, underscoring the merits and potential uses of each technique. These analyses offer crucial insights to refine biochar manufacturing processes and select the method depending on distinct needs and goals.

### 2.2. Characterization of Biomass and Biochar Samples

To assess the impact of various preparation methods (Section 2.1) on the chemical properties, the biomass and biochar samples underwent characterization. The evaluation process involved proximate characterization according to the established ASTM D1762-84 standards, which determined the volatile matter, ash content, and moisture of both the biomass and biochar samples [62,63]. To ascertain the moisture content, the specimens were oven-dried at a temperature of 105 °C over 2 h. The detailed procedure for this method is described as follows.

### 2.2.1. Proximate Analysis

To evaluate the impact of different pyrolysis techniques on the physical and chemical characteristics, biomass and biochar samples underwent characterization. The volatile matter (VM) content of the biochar samples was ascertained following the ASTM D 1762-84 [62,63] standard. This involved a sequential muffle procedure to measure weight loss and mass balance. The percentage of volatile matter was calculated by subtracting the weight loss due to moisture from the total weight loss of the sample using the equations given below; the data are presented as the mean ± standard error of the mean.

$$\text{Volatile matter (\%)} = (\text{Mbiochar or stalk} - \text{MCC/Mbiochar or stalk}) \times 100 \qquad (2)$$

Here, Mbiochar or stalk represents initial dry biochar or stalk mass, and MCC is biochar or stalk mass remaining after heating.

Following the ASTM D 1762-84 [28,63] standard, the carbonized biochar or stalk residue generated during the volatile matter was used to estimate the ash content of the PPS and biochar samples. The equation used to calculate the ash content was as follows:

$$\text{Biochar or Stalk ash (\%)} = (\text{Mash/Mbiochar or stalk}) \times 100 \qquad (3)$$

In this equation, Mash represents the ash's dry mass remaining after the dry combustion of both (carbonized biochar or stalk), while Mbiochar represents biochar or stalk's initial dry mass.

The fixed carbon content was evaluated using the following equation [64]:

$$\text{Fixed carbon (FC) (\%)} = (100 - \%\text{VM} - \%\text{Ash}). \qquad (4)$$

### 2.2.2. Ultimate Analysis

The total carbon (C) and nitrogen (N) content in the PPS and biochar samples were identified using dry combustion (TCTN analyzer, Model: PRIMACS-SNC100 and ICP–AES). Based on the outcomes of the total C and N evaluation, the C/N ratio was computed. Phosphorus (P), potassium (K), sulfur (S), iron (Fe), zinc (Zn), copper (Cu), boron (B), and manganese (Mn) were isolated using distinct procedures. Sodium bicarbonate was used for extracting P [65], ammonium acetate for K [66,67], 0.15% calcium chloride for S [68], hot water for B [69], and diethylene triamine penta-acetic acid (DTPA) was utilized for the extraction of Zn, Cu, Fe, and Mn [70,71]. P was determined using the colorimetric method, while K was measured using an atomic absorption spectrophotometer (AAS). The analysis of S, Zn, Cu, Fe, Mn, and B was conducted using Inductively Coupled Plasma Atomic Emission Spectroscopy (ICP–AES) in the Charles Renard Analytical Laboratory (CRAL) Lab at ICRISAT. The data are presented as the mean ± standard error of the mean.

### 2.2.3. Fuel Properties

The elemental composition of hydrogen (H) and oxygen (O) in the PPS was estimated using empirical correlation Equations (5) and (6), as proposed by [4], based on the results from the proximate analysis; the data are presented as the mean ± standard error of the mean.

$$\text{H (\%)} = 0.052 \times \text{FC} + 0.062 \times \text{VM} \qquad (5)$$

$$\text{O (\%)} = 0.304 \times \text{FC} + 0.476 \times \text{VM} \qquad (6)$$

Here, FC stands for the proportion of fixed carbon, and VM for the proportion of volatile matter in the PPS.

To ascertain the calorific or higher heating value (HHV) of the PPS and biochar specimens, Equation (7) was employed.

$$\text{HHV (MJ kg}^{-1}) = 0.3536 \times \text{FC} + 0.1559 \times \text{VM} - 0.0078 \times \text{Ash} \qquad (7)$$

In this equation, FC stands for the fixed carbon content %, VM for the volatile matter content percentage, and Ash for the ash content percentage in the PPS and biochar [4].

Equations (8)–(13) were used to compute various fuel parameters, utilizing the information from the product yield and proximate analysis of the biochar [72,73]:

$$\text{Energy densification (Ed)} = \text{HHV of dried biochar}/\text{HHV of dried PPS} \tag{8}$$

$$\text{Energy retention efficiency (ERE) (\%)} = \text{Ed} \times \text{biochar yield} \tag{9}$$

$$\text{HHV improvement (HHVi)} = (\text{HHV of dried biochar} - \text{HHV of dried PPS})/\text{HHV of dried PPS} \tag{10}$$

$$\text{Fixed carbon densification (FCd)} = \text{FC of dried biochar}/\text{FC of dried PPS} \tag{11}$$

$$\text{Fixed carbon recovery efficiency (FCre) (\%)} = \text{FCd} \times \text{Biochar yield} \tag{12}$$

$$\text{Fuel ratio (Fr)} = \text{Fixed carbon of biochar}/\text{Volatile matter of biochar} \tag{13}$$

### 2.2.4. Biochar Stability

The stability of biochar was assessed using the alpha method, based on its H/C, O/C, and VM properties [74]. Equations (14) and (15) were used to approximate the H/C and O/C atomic ratios [28]:

$$\text{H/C} = 0.397 \times (\text{VM/FC}) + 0.251 \tag{14}$$

$$\text{O/C} = 0.188 \times (\text{VM/FC}) + 0.035 \tag{15}$$

In this context, FC represents the fixed carbon content percentage, VM stands for the percentage of volatile matter content, H/C indicates the hydrogen–carbon ratio, and O/C signifies the oxygen–carbon ratio.

### 2.2.5. Carbon Dioxide Reduction Potential

The total potential carbon (TPC) in the biochar was calculated using Equation (16):

$$\text{Total Potential Carbon in biochar (g of C kg}^{-1} \text{ of biochar)} = \text{Total biochar yield (kg of biochar kg}^{-1} \text{ of stalk)} \times \text{Fixed carbon (kg of FC kg}^{-1} \text{ of biochar)} \tag{16}$$

Finally, Equation (17) [75] was used to estimate the carbon dioxide reduction potential ($CO_2$ eq kg$^{-1}$ of biochar):

$$CO_2 \text{ reduction potential} = \text{TPC in biochar (g of C kg}^{-1} \text{ of biochar)} \times (80/100) \times (44/12) \tag{17}$$

### 2.2.6. Scanning Electron Microscopy (SEM) and Fourier Transform Infrared Spectroscopy (FTIR)

The surfaces of the biochar samples were examined with scanning electron microscopy (SEM, specifically the Quanta FEG 250 model, Eindhoven, The Netherlands). Additionally, the samples underwent analysis via Fourier transform infrared spectroscopy (FTIR), using the KBr pellet method, scanning from 4000 to 400 cm$^{-1}$, and conducting 50 scans for each sample within each spectrum [14].

## 3. Results and Discussion

### 3.1. Influence of Different Biochar Production Methods on Biochar Characteristics

#### 3.1.1. Biochar Yield

The conversion behavior of biomass during pyrolysis processes, as well as the yields and characteristics of the resulting biochar, are greatly influenced by various factors [76]. In this study, we prepared biochar using a muffle furnace (BPMF: Biochar prepared in muffle furnace) and a kiln (BPPK: Biochar prepared by pyrolysis in kiln). The production method has a significant impact on biochar output [77]. Every operating parameter had an impact on the yield of biochar [78]. The negative coefficients show that temperature, heating rate, and holding time had a detrimental impact on the biochar yield to optimize the biochar yield; conditions with low temperature, slow heating, and a short retention time were employed. Our study revealed that the yield of biochar from PPS was significantly lower for BPPK compared to BPMF.

The biochar yields significantly varied from 21 to 31% (dry weight basis). The lowest biochar mass yield (21%) was observed in the BPPK method. The yield reduction may be due to the release of more $CO_2$ during pyrolysis in the kiln compared to a closed and controlled environment of the muffle furnace. We recorded a significantly higher biochar yield in BPMF (31%) because of less condensation and volatilization of organic substances present in feedstock [79]. The pyrolysis temperature in the kiln varied between 350 to 450 °C, and in the muffle furnace, it was about 500 °C. The holding time for the kiln was also higher compared to the muffle furnace, which had an adverse effect on the biochar yield. Moreover, there was little oxygen in the kiln, which also decreased the yield of biochar prepared in the kiln. With increased oxygen concentration during pyrolysis, the biochar yield was decreased [5]. The biochar yield was increased by using a low temperature, slow heating rate, and brief holding period [78]. Therefore, the reduction of biochar yield in the kiln as compared to the muffle furnace is the cumulative effect of all the operating parameters of the process.

#### 3.1.2. Proximate Analysis

The proximate analyses in Table 1 and ultimate analyses in Table 2 of biochar as well as biomass are presented. Moisture content in both feedstocks' sun-dried biomass was 7.68 to 7.94%. However, the volatile matter was high, i.e., 77.55 to 78.87%. Among the different methods of biochar preparation, the biochar prepared with a muffle furnace had higher fixed carbon, 84.79 to 86.82%. Fixed carbon in the biochar, viz., muffle furnace biochar and drum biochar, was 84.79 to 86.82% and 65.68 to 66.75%, respectively. High ash content was recorded in BPPK (22.45 to 23.40%), and the lowest was in BPMF (4.25 to 4.84%). The significant conversion to pyrogenic vapor during heat treatment and the poor output of biochar are shown by the high quantity of VM and low amount of ash content [76]. Biochars possessing reduced ash content exhibit heightened resistance to wind-induced dispersion, rendering them optimal for transport and seamless integration into soil systems [76].

**Table 1.** Proximate analysis of pigeonpea stalks (biomass) and biochar produced by different methods.

| S.No | Parameters | Pigeonpea Stalks (Biomass) | Biochar Prepared in Muffle Furnace (BPMF) | Biochar Prepared in A Pyrolysis Kiln (BPPK) |
|------|------------|----------------------------|-------------------------------------------|---------------------------------------------|
| 1 | Moisture content (wt %, wb) | 7.68 ± 0.26 | 6.12 ± 0.42 | 6.88 ± 0.15 |
| 2 | Volatile matter (wt %. db) | 77.55 ± 1.37 | 4.56 ± 0.54 | 17.56 ± 0.77 |
| 3 | Fixed carbon (wt %. db) | 14.21 ± 0.90 | 84.79 ± 2.03 | 65.68 ± 1.07 |
| 4 | Ash (wt %. db) | 1.96 ± 0.31 | 4.25 ± 0.59 | 22.45 ± 0.95 |

Values are expressed as mean ± standard error of the mean.

**Table 2.** Ultimate analysis of pigeon pea stalks (biomass) and biochar prepared in muffle furnace (BPMF) and biochar prepared in a pyrolysis kiln (BPPK).

| S.No | Parameter | Pigeonpea Stalks (Biomass) | Biochar Prepared in Muffle Furnace (BPMF) | Biochar Prepared in A Pyrolysis Kiln (BPPK) |
|------|-----------|----------------------------|-------------------------------------------|---------------------------------------------|
| 1 | C (wt. %) | $40.93 \pm 0.85$ | $45.25 \pm 1.21$ | $62.41 \pm 1.25$ |
| 2 | N (wt. %) | $1.66 \pm 0.26$ | $2.12 \pm 0.34$ | $1.18 \pm 0.19$ |
| 3 | C/N | 24.66 | 21.34 | 52.88 |

Note: C/N = Carbon and nitrogen ratios. Values are expressed as mean $\pm$ standard error of the mean.

### 3.1.3. Ultimate Analysis

Carbon, hydrogen, nitrogen, sulfur, oxygen, and elemental ratios of both the feedstocks relating to the qualitative traits were studied and represented in Table 1. The C, N, and C/N of PPS were found to be 40.93%, 1.66%, and 24.66%, respectively, and it was the lowest compared to biochar samples. The C, N, and C/N of biochar prepared in a muffle furnace were 45.25–46.46%, 2.12–2.46%, and 21.34%, respectively. The C, N, and C/N of biochar prepared in the kiln were 62.41–63.66%, 1.18–1.37%, and 52.88%, respectively. The composition of the material used as feedstock is important in determining the characteristics and durability of biochar. Our research shows that, even though biochar has a higher ratio of carbon to nitrogen, it is still resistant to microbial decay. This suggests that, when used to amend soil, biochar has little effect on the immobilization of nitrogen in soil [77].

### 3.1.4. Elemental Composition

The elemental composition of biochar prepared using different methods is illustrated in Figure 1. The organic carbon content ranged from 7.54% to 13.27%, depending on the biochar production method. Specifically, BPMF exhibited an organic carbon content of 7.54%, while BPPK had an organic carbon content of 13.27% (average values were considered).

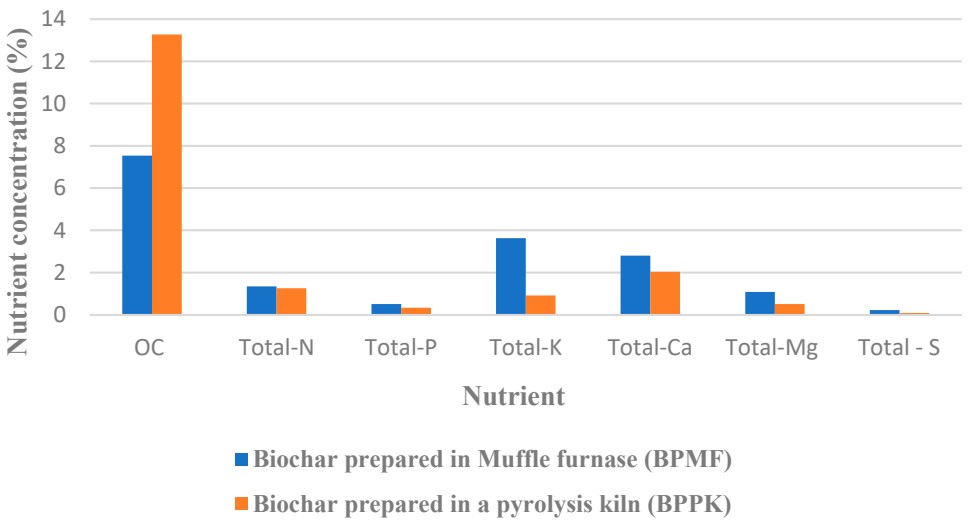

**Figure 1.** Organic carbon (OC) and primary and secondary nutrient composition of biochar prepared in a muffle furnace (BPMF) and biochar prepared in a pyrolysis kiln (BPPK).

Regarding nutrient concentration, BPMF displayed a higher concentration of potassium (K) compared to all other nutrients, whereas sulfur (S) had the lowest concentration. Notably, K exhibited greater variation in nutrient concentration between the two methods, while nitrogen (N) demonstrated relatively low variation.

Among different methods of biochar preparation, BPMF recorded a higher micronutrient composition than BPPK. The micronutrient composition ranged from 33.6 (Cu) to 12,100 (Fe) ppm. The micronutrient composition was 115.3 ppm (Zn), 141.0 (B), 1210 ppm

(Fe), 33.68 ppm (Cu), 305.00 ppm (Mn), and 1798 ppm (Na) in BPMF, while the values in BPPK were 87.50 ppm (Zn), 70.80 ppm (B), 4697.5 ppm (Fe), 40.00 ppm (Cu), 100 ppm (Mn), and 1370.00 ppm (Na). Among the micronutrients, Fe was the highest (1210 ppm), while the lowest was Cu (33.6 ppm) (Figure 2).

The analysis revealed that both methods of biochar production generally resulted in higher cation concentrations compared to anions. These findings align with a previous study conducted on castor stalk biochar [57], which highlighted the significant influence of stalk nutrient concentrations and kiln temperature on the biochar's nutrient content. Specifically, it was determined that a charring temperature exceeding 760 °C was required for potassium (K) vaporization and 800 °C for phosphorus (P) vaporization [80,81]. In the current study, the selected upper limit for the charring temperature was 450 to 500 °C. Hence, it is reasonable to believe that the original nutrient levels in the stalk, coupled with decreased volatilization losses, played a role in the efficient preservation of nutrients within the biochar [32].

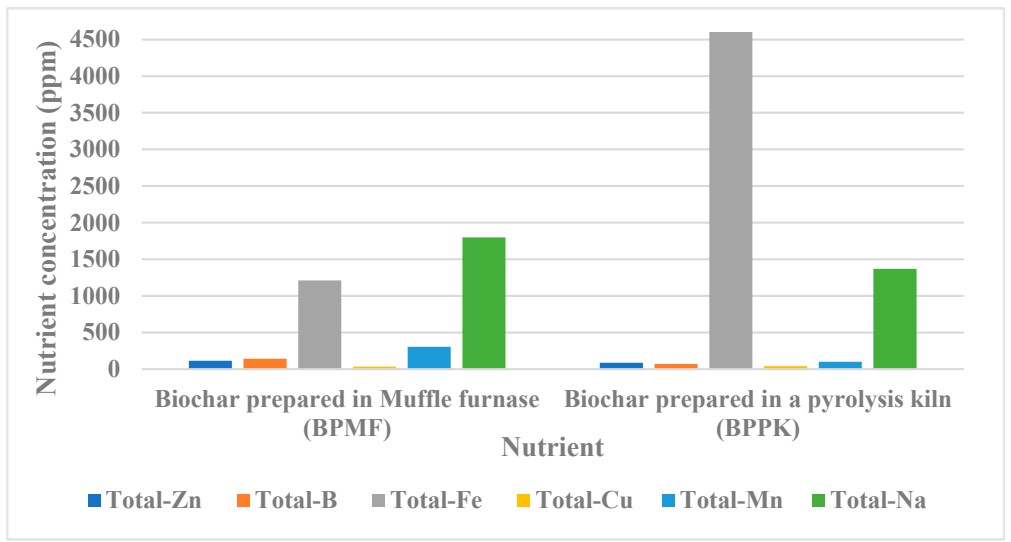

**Figure 2.** Effect of different biochar production methods on micronutrient composition (ppm).

### 3.2. Energy Use and Fuel Properties of Biochar from Different Production Techniques

Table 3 provides the values for energy-related properties. These properties include fuel ratio (Fr), higher heating value (HHV), energy retention efficiency (ERE), densification of energy (Ed), higher heating value improvement (HHVi), fixed carbon densification (FCd), and fixed carbon recovery efficiency (FCre), which have been calculated or estimated. The parameters discussed here serve as the foundation for assessing the qualitative characteristics of biochar as a source of energy. The energy characteristics of biochar are enhanced through chemical dehydration and decarboxylation reactions, which result in the release of water vapor ($H_2O$) and carbon dioxide ($CO_2$) [72].

Energetic retention efficiency (ERE) is a crucial characteristic to consider when evaluating the impact of the production process for biochar as a solid fuel substitute. The energetic retention efficiency quantifies how much PPS energy was retained in the biochar. The energetic retention efficiency was greater in BPMF (55.58–57.42%) than in BPPK (31.66–33.60%), as biochar yield was higher in BPMF (31–33%). The energetic retention efficiency was low in BPPK (32.24%) because of the decreased yield of biochar (21–23%). These study results suggest that the BPMF method of biochar production is efficient for the production of energy-rich biochar.

**Table 3.** Effect of biochar prepared in muffle furnace (BPMF) and biochar prepared in a pyrolysis kiln (BPPK) methods on energy use.

| S.No | Parameter | Biochar Prepared in Muffle Furnace (BPMF) | Biochar Prepared in A Pyrolysis Kiln (BPPK) |
|------|-----------|-------------------------------------------|---------------------------------------------|
| 1 | HHV (MJ kg$^{-1}$) | 30.66 ± 1.90 | 25.79 ± 1.50 |
| 2 | Energy densification (Ed) | 1.80 ± 0.29 | 1.50 ± 0.18 |
| 3 | Energy retention efficiency (ERE) (%) | 55.58 ± 1.84 | 31.66 ± 1.94 |
| 4 | HHV improvement (HHVi) | 0.79 ± 0.09 | 0.51 ± 0.07 |
| 5 | Fixed carbon densification (FCd) | 5.86 ± 0.97 | 4.70 ± 0.93 |
| 6 | Fixed carbon recovery efficiency (FCRE) (%) | 181.66 ± 2.96 | 98.70 ± 1.76 |
| 7 | Fuel ratio (Fr) | 18.59 ± 3.43 | 3.00 ± 0.65 |

Values are expressed as mean ± standard error of the mean.

Energy densification (Ed) increased in BPMF (1.80 to 2.09). Energy densification (Ed) in biochar has been observed in various crop residues [82,83], indicating its potential as a concentrated energy source. An important metric for assessing the fuel value of biochar is its higher heating value (HHV), which quantifies the energy content of the material. The HHV signifies the energy liberated per unit weight or volume of the fuel following total combustion. This includes the energy emitted during burning and the energy related to turning water into vapor [74].

The Higher Heating Value (HHV) of biochar is between 25.79–32.56 MJ kg$^{-1}$, as detailed in Table 4. Interestingly, BPMF samples exhibited a higher HHV (30.66 to 32.56 MJ kg$^{-1}$) in comparison to the BPPK samples. This discrepancy in HHV is likely due to variations in ash content proportions [74]. Moreover, BPMF's HHV (30.66 to 32.56 MJ kg$^{-1}$) was marginally greater than Dimethyl ether's (29 MJ kg$^{-1}$), while BPPK's HHV (25.79 to 27.29 MJ kg$^{-1}$) trailed behind Dimethyl ether's value [84]. The elevation in HHV is believed to result from the evaporation of low-energy constituents like oxygen, hydrogen, and nitrogen, leaving the high-energy carbon component more concentrated. The increase in HHV may be due to the evaporation of low-energy constituents such as oxygen, hydrogen, and nitrogen. This process leaves the high-energy carbon component more concentrated, resulting in elevated HHV levels [73].

**Table 4.** Effect of biochar prepared in muffle furnace (BPMF) and biochar prepared in a pyrolysis kiln (BPPK) methods on H, O, O/C, and H/C atomic ratios, total potential carbon, and $CO_2$ reduction potential of PPS biochar.

| S.No. | Parameter | Biomass | Biochar Prepared in Muffle Furnace (BPMF) | Biochar Prepared in A Pyrolysis Kiln (BPPK) |
|-------|-----------|---------|-------------------------------------------|---------------------------------------------|
| 1 | H (wt. %) | 6.40 ± 0.35 | 5.34 ± 0.74 | 5.27 ± 0.95 |
| 2 | O (wt. %) | 47.58 ± 1.05 | 31.80 ± 1.69 | 33.05 ± 1.73 |
| 3 | H/C (wt. %) | 0.15 | 0.11 | 0.08 |
| 4 | O/C (wt. %) | 1.16 | 0.70 | 0.53 |
| 5 | TPC in biochar (g of C kg$^{-1}$ of biochar) | - | 262.8 ± 1.50 | 138.00 ± 0.69 |
| 6 | $CO_2$ reduction potential ($CO_2$ eq kg$^{-1}$) | - | 77.17 ± 2.51 | 40.46 ± 1.44 |

Values are expressed as mean ± standard error of the mean.

The higher fixed carbon recovery efficiency (181.66 to 184.62%), fixed carbon densification (5.86 to 6.83%), energy density (1.80 to 2.09%), energetic retention efficiency (55.58 to 57.42%), fuel ratio (18.95 to 22.38%), and HHV (30.66 to 32.56 MJ kg$^{-1}$) indicate that the biochar produced in BPMF from pigeonpea stalks can be used as an alternative solid

fuel [75]. The BPPK reported lower fixed carbon recovery efficiency (98.70 to 100.46%), fixed carbon densification (4.70 to 5.63%), energy density (1.50 to 1.68%), energetic retention efficiency (31.66 to 33.6%), fuel ratio (3 to 3.65%), and HHV (25.79 to 27.29 MJ kg$^{-1}$) [73].

### 3.3. Biochar Stability Analysis

The biomass samples prepared in a muffle furnace (BPMF) and a pyrolysis kiln (BPPK) had hydrogen content (H wt. %) ranging between 5 and 6.5%. In contrast, the oxygen content (O wt. %) was significantly different, with BPMF having almost 50% oxygen and BPPK having around 30% oxygen. Table 4 showcases a uniform ratio decline for all biochar, which implies a reduction in degradable polar components [44,62,83]. This decline can be linked to condensation and aromatization reactions that become more pronounced at elevated temperatures [64]. The O/C and H/C atomic ratios for biochar produced using both methods were 0.08 to 0.11 and 0.53 to 0.70, respectively. Notably, the BPPK samples recorded lower values. A declining H/C atomic ratio points towards heightened structural stability in pigeonpea biochar, resulting from enhanced aromatization [79]. This change is likely due to hydrogen removal via dehydration and dehydrogenation processes and weaker hydrogen bond disruption during biochar formation [77]. The reduced O/C atomic ratio also signals greater carbonization, achieved by removing oxygen through dehydration and decarboxylation reactions during the transformation process [63,79].

Such attributes amplify the biochar's chemical resilience, rendering it less susceptible to microbial breakdown [41]. This resilience ensures that the biochar remains in the soil for lengthy durations, potentially spanning centuries [81]. Earlier research has posited that biochar created at elevated pyrolysis temperatures boasts greater aromatic character and reduced H/C and O/C atomic ratios than its counterparts produced at cooler temperatures [58]. The data underscore the effective transformation of PPS into biochar in our study. As the temperature rose across all three batches, the H/C and O/C atomic ratios diminished, pointing to refined fuel qualities that are on par with lignite and sub-bituminous coal, possessing higher heat yields.

### 3.4. CO$_2$ Reduction Potential of Biochar from Different Production Methods

The amount of TPC (Equation (16)) and CO$_2$ reduction potential (Equation (17)) of the biochar produced using different methods of preparation ranged from 162.75 to 301.41 g of C kg$^{-1}$ of biochar (Table 4). The BPMF recorded the highest TPC (138.00 to 264.30 g of C kg$^{-1}$ of biochar) and CO$_2$ reduction potential (40.46 to 79.68 CO$_2$ eq kg$^{-1}$). The highest TPC (262.8 to 264.3 g of C kg$^{-1}$ of biochar) and CO$_2$ reduction potential (77.17 to 79.68 CO$_2$ eq kg$^{-1}$) was recorded in BPMF, while the lowest TPC was recorded in BPPK (138.00–138.60 g of C kg$^{-1}$ of biochar) and CO$_2$ reduction potential (40.46–41.90 CO$_2$ eq kg$^{-1}$) was recorded in BPPK. Both types of biochar (BPMF and BPPK) have the potential to reduce CO$_2$, with the biochar prepared in BPMF having a more significant CO$_2$ reduction potential (CO$_2$ eq kg$^{-1}$) of 77.17 to 79.68 CO$_2$ eq kg$^{-1}$ compared to BPPK's 40.46–41.90 CO$_2$ eq kg$^{-1}$.

Using this method to transform pigeonpea stalks into biochar produced a more resilient carbon form (biochar), capable of resisting microbial breakdown. As a result, it offers a potential avenue for capturing and storing atmospheric CO$_2$ in the soil.

If crop waste containing unstable carbon is left to decompose naturally or burned in situ, it quickly converts into carbon dioxide. However, research has shown that pigeonpea biochar can help mitigate this process. Its high organic carbon content (ranging from 734.00 to 734.17 g kg$^{-1}$) makes it a potential agent for long-term carbon storage in soil, which can serve as a climate change mitigation option. This is because a significant amount of atmospheric carbon dioxide (approximately 87.47–87.81 CO$_2$ eq kg$^{-1}$) can be converted into a more stable form of carbon that is resistant to degradation and remains in the soil for a longer period [82].

In 2016–2017, according to data from India's Ministry of Agriculture & Farmers Welfare, the country produced 18.53 million tons of pigeonpea annually from an area of 5.34 million hectares, with an average yield of 0.913 tons per hectare. The residue from this crop had

a ratio of 3.8 tons per hectare. Current findings suggest that the biochar potential from pigeonpea stalks in India ranges between 5.1 and 6.11 million tons yearly. This could leads to a primary carbon total of 2.55–2.57 million tons each year and a potential $CO_2$ reduction of 7.49–7.76 million carbon equivalent tons annually. When biochar is created using BPMF, it has a yield of 31–33% and a fixed carbon content of 84.79–86.82% in pigeonpea stalk (PPS) biochar. Applying this biochar to farmland can trap about 3.31–3.41 million tons of carbon every year, showcasing its capability as a carbon capture method.

### 3.5. Scanning Electron Microscopy (SEM) and Fourier Transform Infrared Spectroscopy (FTIR)

The electron micrograph of biochar samples revealed clearly that biochar prepared using a muffle furnace (BPMF) is more porous compared to biochar prepared using a kiln (BPPK), having micropores; however, each biochar shows crystalline depositions near and in the micropores (Figure 3). The pore sizes of BPMF and BPPK vary from 0.72 μm to 1.72 μm. Biochar production techniques and characterization of biochar propose that the surface area and porosity are influenced by the production process and temperature [85,86]. Additionally, the expansion of temperature during biochar test upgrades may result in significant improvement in the pore characteristics of biochar.

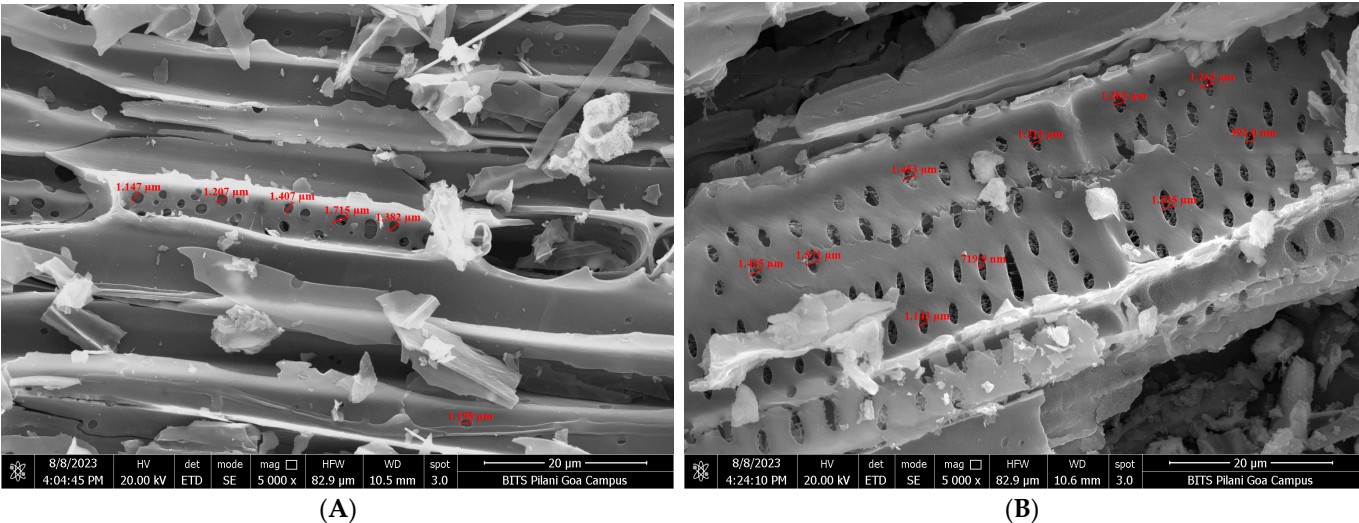

**Figure 3.** Electron micrograph of biochar prepared in kiln (**A**) and biochar prepared in muffle furnace (**B**).

As per the FTIR analysis of biochar prepared in the muffle furnace (BPMF) and kiln (BPPK), functional groups, viz., carbonyl (C=O), C=C, and aromatic C-H groups, are present. The transmittance of the functional groups is more intense in BPMF compared to BPPK. The vibrations associated with the S-S linkage range from 500 to 400 cm$^{-1}$. The vibration that stretches the disulfide is faint. Disulfide S-S stretching vibrations are shown by the FTIR peak between 490.6 and 432.9 cm$^{-1}$. The results of FTIR are depicted in Figures 4 and 5 [14,86].

A study evaluated biochar produced through the torrefaction of sawdust and peat and through hydrothermal carbonization across various temperatures [20]. Multiple techniques, including elemental analysis, SEM, and FTIR, were employed to assess the biochar properties. Results indicated that hydrothermal carbonization has a more pronounced impact on the carbonification level of the original biomass compared to torrefaction. Biochar derived from peat via this method exhibited a more resilient structure, with features resembling asphaltene fragments and an intricate system of linkages. The properties of biochar vary based on the feedstock's origin and the temperatures used during production. The study underscored that production temperature notably influences the yield, pH, stability, and volatile content of biochar. Conversely, the source of the feedstock primarily impacts

characteristics like carbon content, cation exchange capacity (CEC), fixed carbon, carbon storage potential, mineral levels, and ash content [86].

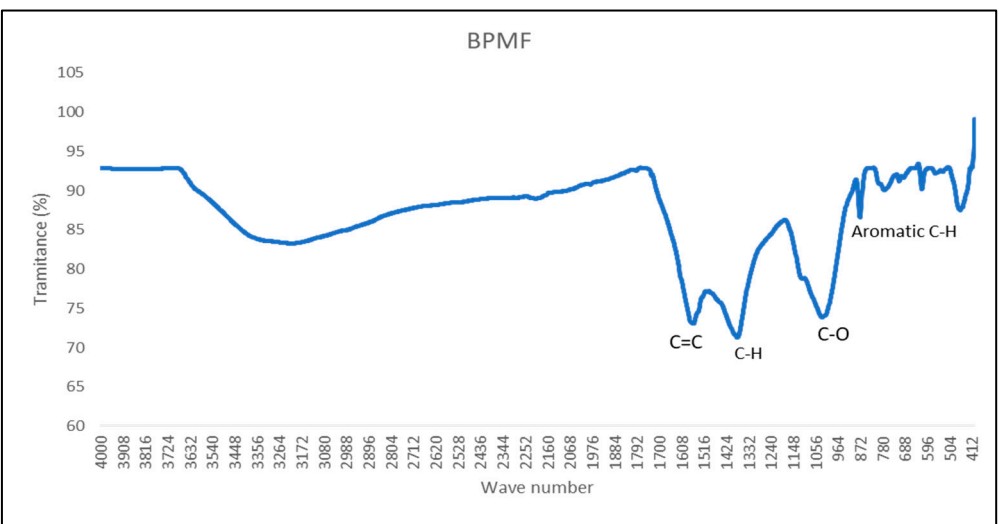

**Figure 4.** FTIR of biochar prepared in muffle furnace.

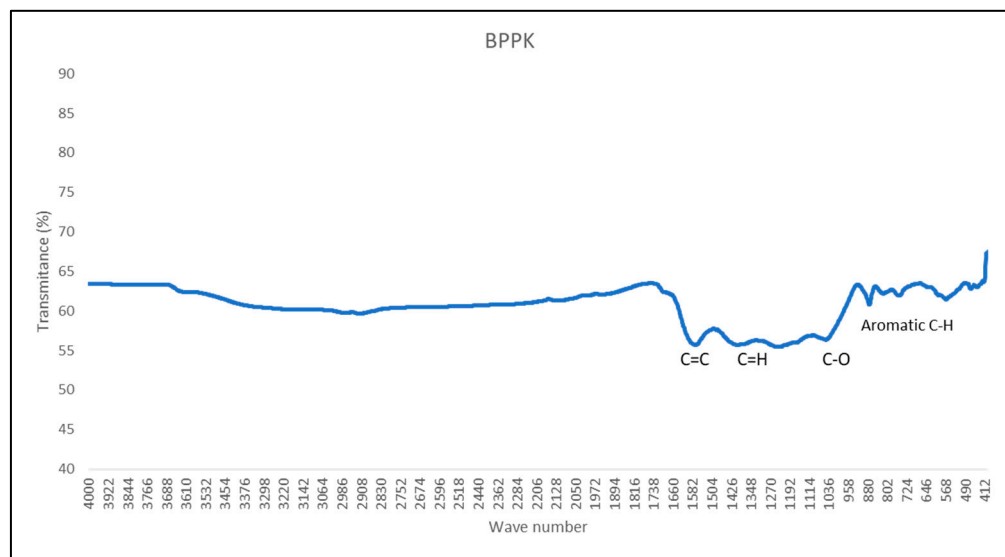

**Figure 5.** FTIR of biochar prepared in biochar kiln.

## 4. Conclusions

This study emphasizes the advantages of muffle furnace technology and the potential of pigeonpea stalks for the synthesis of biochar. Here, we compared two important biochar production methods, i.e., muffle furnace and kiln. The biochar produced from muffle furnace showcases substantial fixed carbon content (over 80%) and a notable energy density (1.77 to 2.06 MJ kg$^{-1}$), making it a valuable option for both soil enhancement and renewable fuel. Comparing different biochar production methods reveals distinct advantages of muffle furnace over the kiln, and BPMF stands out, with its remarkable fixed carbon recovery efficiency, energy density, and retention. Expanding this approach to other crop residues offers exciting possibilities, and evaluating biochar's practical economic and environmental impact further cements its potential. By adopting a muffle furnace biochar production method in place of the existing kiln method, we can restore soils, curb carbon emissions, and diversify energy sources. Converting pigeon pea stalks into biochar through a muffle furnace will make our agriculture more resilient, helping to combat climate change.

**Author Contributions:** Conceptualization, N.V.K.; methodology, N.V.K.; formal analysis, N.V.K.; investigation, N.V.K.; resources, G.L.S. and C.S.R.; data curation, N.V.K.; writing—original manuscript, N.V.K.; writing—review and editing, N.V.K., G.L.S., C.S.R., A.S., P.J.K. and R.P.; supervision, T.R.P., S.T., R.K. and B.V. All authors have read and agreed to the published version of the manuscript.

**Funding:** This research received no external funding.

**Institutional Review Board Statement:** Not applicable.

**Informed Consent Statement:** Not applicable.

**Data Availability Statement:** The data are available on request from the corresponding author for reasonable reasons.

**Acknowledgments:** We acknowledge the support of Central Sophisticated Instrumentation Facility, BITS-Pilani, K.K. Birla Goa campus for conducting SEM and FTIR analyses.

**Conflicts of Interest:** The authors declare no conflict of interest.

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
