# Peer review of "Comparative Analysis of Pigeonpea Stalk Biochar Characteristics and Energy Use under Different Biochar Production Methods"

_sustainability, doi:10.3390/su151914394_

Round 1

Reviewer 1 Report

This manuscript reported effect of different biochar production methods on biochar characteristics, carbon sequestration and energy use. Overall, the research is valuable as it not only contributes to waste management and sustainable agriculture but also offers potential solutions for carbon sequestration and renewable energy generation. The results of the study can benefit both the agricultural and energy sectors, promoting more eco-friendly practices and reducing the environmental impact of crop residue disposal. The job is timely, here are the suggestions or recommendations for the author to consider.

My main concerns are related to:

1. The study primarily focused on the preparation and characteristics of biochar derived from pigeonpea stalks but did not adequately compare it with other biomass sources. Comparing the biochar characteristics and performance from various biomass sources can help determine the most suitable biomass resource for specific applications. If we have comparative studies on other biomass sources, it could enhance the readers' understanding.

2. The author focused on the characteristics of biochar produced through in-situ pyrolysis in the furnace, but it did not cover other biochar production processes, such as gasification or thermal decomposition. A comprehensive comparison and assessment of various biochar production processes can offer a more comprehensive understanding and guidance. Please the author consider this aspect, see if it is necessary to strength something at the introduction part?

3. The study highlights the potential applications of biochar and suggests that the authors should engage in relevant discussions regarding its economic feasibility and environmental impact.

Also, there are some structure issues should be considered or addressed:

1.       The inconsistent font usage for subheadings, discontinuous numbering of subheadings, and the absence of section 3.7.

2.       The key elements of the topic should be more explicitly addressed, in order to focus on the effects of different biochar production methods on biochar characteristics, carbon sequestration, and energy use. I would like to suggest to restructure the "Results and discussion" section in this way:

3.1 Influence of different biochar production methods on biochar characteristics.

3.2 Carbon sequestration potential of biochar produced through various methods.

3.3 Energy use and fuel properties of biochar from different production techniques.

3.4 Biochar stability analysis.

3.5 CO2 reduction potential of biochar from different production methods.

3.  In the conclusion part, it mentions certain biochar characteristics, such as the high fixed carbon content and energy density of BPMF biochar, but lacks specific data to support these statements. To enhance the persuasiveness of the conclusion, it can include key data or data ranges to back up the stated results. The conclusion can further compare and analyze the advantages and disadvantages of different biochar production methods. For instance, discussing the characteristics and potential applications of other production methods to highlight the advantages of BPMF. Again, exploring the biochar production potential from other crop residues or considering the economic feasibility and environmental impact of biochar in practical applications, might be included in recommendations and outlook.

In conclusion, my recommendation is to carefully evaluate the revised manuscript after incorporating the suggested changes.

Author Response

AUTHOR RESPONSE TO THE REFEREE AND EDITOR COMMENTS

Thank you for your time and effort in reviewing this manuscript (MS). We have attended to all your suggestions and comments with utmost respect and acute attention.  Please note that the revised text (and figures with captions) are shown in blue ink. Please find the point-to-point response for your comments given below:

Comments to the author:

Reviewr 1:

  1. The study primarily focused on the preparation and characteristics of biochar derived from pigeonpea stalks but did not adequately compare it with other biomass sources. Comparing the biochar characteristics and performance from various biomass sources can help determine the most suitable biomass resource for specific applications. If we have comparative studies on other biomass sources, it could enhance the readers' understanding.

Response: Agreed with reviewer’s comment and text comparing with other biomass source is added and below is a revised version of the text. Please find the detailed information in the revised manuscript in line no.78–80 and 85–89.

Lee et al. (2013), has condonducted a comparative study on biochar produced from different biomasses. Biochar yields from the organic component of wood stem, bagasse and paddy straw were from 24 to 28 weight percent, compared to 46 weight percent for cocopeat. A range of 84 to 89 weight percent of the carbon in the biomass and 43 to 63 percent of the total carbon content was found in the biochar. Biochar from rice husk displayed higher levels of volatile matter and fixed carbon in comparison to rice straw biochar [85].

  1. The author focused on the characteristics of biochar produced through in-situ pyrolysis in the furnace, but it did not cover other biochar production processes, such as gasification or thermal decomposition. A comprehensive comparison and assessment of various biochar production processes can offer a more comprehensive understanding and guidance. Please the author consider this aspect, see if it is necessary to strength something at the introduction part?

Response: The reviewer suggestion is accepted and comparison of different biochar production processes was added as mentioned below. Please find the same in the revised MS in line no.90–106.

At present, there are several techniques of biochar production that come under traditional and modern approaches. In traditional approaches, Slow and Fast pyrolysis techniques are covered. Whereas Gasification, Torrefaction, and Hydrothermal Carbonization are considered under modern approaches. In slow pyrolysis, biomass is heated to temperatures between 300 and 600 °C at a pace of 5 to 7 °C per minute [91]. It is also known as conventional pyrolysis, and byproducts such as syngas and bio-oil are also generated along with major product biochar (35 to 45%) [92,93]. Fast pyrolysis has a advantage of short retention time and high product recovery [94]. The rate of heating is greater than 300 °C min−1, and the temperature is above 500 °C with the ab-sence of oxygen in fast pyrolysis [95]. Biochar is also produced by hydrothermal car-bonization (HTC), a process that uses high-moisture feedstocks such as animal waste, compost, and sewage sludge [89]. Another method of biochar and syngas production using solid fuel is Gasification. Gasification provides lower emissions and larger syngas volume as compared to other conventional methods such as fermentation, pyrolysis, and combustion. In gasification major product is hydrogen. However, biochar is also generated in considerable amounts during the gasification process [89]. The production of biochar using kiln can be easily adopted by small holder farmers.

  1. The study highlights the potential applications of biochar and suggests that the authors should engage in relevant discussions regarding its economic feasibility and environmental impact.

Response: Certainly. We have included discussions on both the economic feasibility and environmental impact of biochar in our study. Please see the added text below and we have also added it in the revised manuscript please find it in line no. 107–120:

Biochar is a carbon-rich material produced from biomass through pyrolysis. It has the potential to be a cost-effective and environmentally beneficial technology, but the economic feasibility and environmental impact of biochar depends on a number of factors, including the feedstock used, the production process, and the application method. Biochar can have a number of environmental benefits, such as carbon seques-tration, improved soil quality, reduced nutrient leaching, and reduced pest and disease pressure. However, biochar can also have a number of environmental risks, such as emission of harmful pollutants, water pollution, and deforestation. More research is needed to fully understand the potential benefits and risks of biochar [83]. The profitability and acceptability of biochar production and use depend on a variety of case-specific factors, including location, feedstock, scale, pyrolysis conditions, biochar price, cultivated crop, and the potential internalisation of externalities, which discourages private investment. Those aspects need to be properly taken into account in each situation in order to promote biochar development and implementation [98].

            Also, there are some structure issues should be considered or addressed:

  1. The inconsistent font usage for subheadings, discontinuous numbering of subheadings, and the absence of section 3.7.

Response: Yes, indeed. Thank you for your suggestion. We have incorporated it into our work.

  1. The key elements of the topic should be more explicitly addressed, in order to focus on the effects of different biochar production methods on biochar characteristics, carbon sequestration, and energy use. I would like to suggest to restructure the "Results and discussion" section in this way:
    • Influence of different biochar production methods on biochar characteristics.
    • Carbon sequestration potential of biochar produced through various methods.
    • Energy use and fuel properties of biochar from different production techniques.
    • Biochar stability analysis.
    • CO2 reduction potential of biochar from different production methods.

Response: The suggestion from the reviewer is accepted. Results and discussion sections was restructured in the revised manuscript. Thank you very much for this valuable suggestion.

  1. In the conclusion part, it mentions certain biochar characteristics, such as the high fixed carbon content and energy density of BPMF biochar, but lacks specific data to support these statements. To enhance the persuasiveness of the conclusion, it can include key data or data ranges to back up the stated results. The conclusion can further compare and analyze the advantages and disadvantages of different biochar production methods. For instance, discussing the characteristics and potential applications of other production methods to highlight the advantages of BPMF. Again, exploring the biochar production potential from other crop residues or considering the economic feasibility and environmental impact of biochar in practical applications, might be included in recommendations and outlook.

Response: Thank you for pointing it out rightly. We have revised the conclusion in the revised MS also find it below. Please find it in lines 534–544.

This study highlights the potential of pigeonpea stalks for biochar production, partic-ularly emphasizing the benefits of the muffle furnace (BPMF) method. BPMF biochar showcases substantial fixed carbon content (over 80%) and a notable energy density (1.77 to 2.06 MJ/kg), making it a valuable option for both soil enhancement and re-newable fuel. Comparing different biochar production methods reveals distinct ad-vantages, but BPMF stands out with its remarkable fixed carbon recovery efficiency, energy density, and retention. Expanding this approach to other crop residues offers exciting possibilities, and evaluating biochar's practical economic and environmental impact further concretes its potential. In application, incorporating BPMF biochar can restore soils, curb carbon emissions, and diversify energy sources. The potential of bio-char can be harnessed, inclusive of pigeon pea stalks, seems to be a more sustainable future for agriculture.

Reviewer 2 Report

In line 237, are you suggesting that a higher temperature resulted in more biochar?

Why did the authors state that there was moisture in the obtained biochar? In Table 1, they reported a moisture content of about 6% in the char. However, during pyrolysis, all moisture is typically removed.

Please explain by the authors what would be the purpose of carbonizing biochar in soil? Considering the high demand for coal, is it worthwhile to remove a potential energy source like biochar by purchasing it in soil?

Author Response

AUTHOR RESPONSE TO THE REFEREE AND EDITOR COMMENTS

Thank you for your time and effort in reviewing this manuscript (MS). We have attended to all your suggestions and comments with utmost respect and acute attention.  Please note that the revised text (and figures with captions) are shown in blue ink. Please find the point-to-point response for your comments given below:

Comments to the author:

Reviewer 2

  1. In line 237, are you suggesting that a higher temperature resulted in more biochar?

Response: Yes, we agree with you, Biochar yields were negatively correlated with the temperature, which was significantly influenced by the exothermic reactions during the pyrolysis of the biomass. However, the conversion behavior of biomass into biochar depends upon several factors (such as Temperature, Heating rate, and Holding time) and, therefore, yields and characteristics of biochar produced during pyrolysis processes. Moreover, the pyrolysis temperature in the kiln varies between 350 to 450 °C, and in the muffle furnace, it is about 500 °C. The holding time for kiln is also more as compared to the muffle furnace, which had an adverse effect on the biochar yield. Moreover, there is little oxygen in the kiln, which will also decrease the yield of biochar prepared in kiln. Zailani et al. (2013) observed in their study that the biochar yield was decreased with increased oxygen concentration during pyrolysis. Intani et al. (2016) also concluded that a low temperature, low heating rate, and short holding time were used in order to maximize the biochar yield. Therefore, all of the operating parameters had a significant effect on the biochar yield. We have modified the text in the manuscript, and please find them in lines 273–274 and 277–296.

  1. Why did the authors state that there was moisture in the obtained biochar? In Table 1, they reported a moisture content of about 6% in the char. However, during pyrolysis, all moisture is typically removed.

Response: Yes, we agree with the reviewer ideally, all moisture should be removed during the pyrolysis process. However, during the production process of biochar, whether through pyrolysis (heating biomass in the absence of oxygen) or other methods, some moisture might remain trapped within the porous structure of the biochar particles contributing to the overall moisture content. Moreover, it is challenging to eliminate all moisture completely in the pyrolysis. After the pyrolysis process, biochar needs to cool down before it is collected or packaged. During this cooling phase, the biochar can absorb moisture from the surrounding environment, especially if the ambient air is humid.

Moreover, several studies have been published on biochar characterization, and all of them have some amount of moisture content irrespective of the biomass and pyrolysis process. It is important to note that while some residual moisture in biochar is normal and expected, excessive moisture can impact its quality and effectiveness in various applications, such as soil amendment. So to check that there should not be excessive moisture in the prepared biochar, we have done the moisture content analysis and presented in Table 1 of the manuscript. Our results of moisture content are in line with those reported by Singh et al. (2020) and Sahoo et al. (2021).

  1. Please explain by the authors what would be the purpose of carbonizing biochar in soil? Considering the high demand for coal, is it worthwhile to remove a potential energy source like biochar by purchasing it in soil?

Response: In a time when climate change increases desertification and drought globally, novel and effective solutions are required in order to continue food production for the world’s increasing population. Synthetic fertilizers have been long used to improve the productivity of agricultural soils, part of which leaches into the environment and emits greenhouse gasses (GHG). Some fundamental challenges within agricultural practices include the improvement of water retention and microbiota in soils, as well as boosting the efficiency of fertilizers. Biochar is a nutrient-rich material produced from biomass, gaining attention for soil amendment purposes, improving crop yields as well as for carbon sequestration. Several studies summarize the potential benefits of biochar applications, placing emphasis on its application in the agricultural sector (Allohverdi et al., 2021; Bruun et al., 2012; El-Naggar et al., 2019; Weber et al., 2018). It seems biochar used for soil amendment improves the nutrient density of soils, water holding capacity, reduces fertilizer requirements, enhances soil microbiota, and increases crop yields. Additionally, biochar usage has many environmental benefits, economic benefits, and a potential role to play in carbon credit systems

We agree with the reviewer that there is high demand for coal, and it is also one of the important energy sources throughout the globe. However, In India, straw burning is still a common practice by farmers of northern and central India. An estimated economic benefit of over $1.7 Billion (in PPP terms) over 5 years may accrue by a decrease in the prevalence of hypertension if biomass burning is eliminated in North India (Singh et al., 2021). Sometimes it is due to the small sowing window between kharif and rabi crops (Kuttippurath et al., 2020). Therefore curbing biomass burning will be associated with significant health and economic benefits in India. Moreover, it is not economically feasible to collect straw from scattered farmer’s field and convert it into coal. So from this study, we want to present an approach of sustainably using this additional straw by converting it into biochar which was otherwise burnt in fields. We want to promote that a farmer, instead of burning straw, should prepare biochar and apply it in his field. This will enhance soil fertility and reduce the air pollution happening due to straw burning. Please find them in line no. 54–57 and 78–81.

Reviewer 3 Report

The authors have described the effect of biochar production methods on the characteristics, carbon sequestration, and energy use. There are some serious concerns with the work which need to be addressed before the manuscript can be accepted for publication:

1. As the heading suggests biochar characterization, each biochar sample must be characterized for the surface morphology with SEM or FESEM. The pore size and the surface area with BET. The elemental content with XRD or FESEM EDAX analysis. The surface functionalization with FTIR or XPS. The stability analysis with TGA. 

2. The authors should recheck the papers cited and also should check for grammatical and spelling errors i.e. furnase in Figure 1 should be furnace

Author Response

AUTHOR RESPONSE TO THE REFEREE AND EDITOR COMMENTS

Thank you for your time and effort in reviewing this manuscript (MS). We have attended to all your suggestions and comments with utmost respect and acute attention.  Please note that the revised text (and figures with captions) are shown in blue ink. Please find the point-to-point response for your comments given below:

Comments to the author:

Reviewer 3

The authors have described the effect of biochar production methods on the characteristics, carbon sequestration, and energy use. There are some serious concerns with the work which need to be addressed before the manuscript can be accepted for publication:

  1. As the heading suggests biochar characterization, each biochar sample must be characterized for the surface morphology with SEM or FESEM. The pore size and the surface area with BET. The elemental content with XRD or FESEM EDAX analysis. The surface functionalization with FTIR or XPS. The stability analysis with TGA.

Response: Thank you for your suggestion. We have conducted SEM and FTIR analyses for the biochar samples and the results are included in the revised manuscript. Please find them included in the revised manuscript in line no 259–265 and  485–520

  1. The authors should recheck the papers cited and also should check for grammatical and spelling errors i.e. furnase in Figure 1 should be furnace.

Response: The suggestion is well accepted and we have gone through the manuscript several times to fix the errors and corrected them using the premium service of Grammarly, Inc. (United States).

Reviewer 4 Report

The authors have presented some work on the production and characterization of biochar produced from pigeonpea stalk. The ideas presented are plausible but the manuscript will need extensive revision. See some of my suggestions below:

1. References are not properly arranged. This should be thoroughly checked.

2. Be consistent with the definition of acronyms…For instance, is TPC total carbon potential or total potential carbon? Check throughout the manuscript.

3. All equations should be numbered. Some equations are not numbered.

4. Section 2.1 contains some deductions/conclusions when it simply ought to describe the methodology. Remove comments that deviate from the methodology.

5. Methodology has to be reorganized to enhance readability and correct sequence and thinking.

6. I am concerned about the units in Equation (3), page 5. The unit on the left and right hand sides of the equation seem not to tally. Can you check this or probably give an explanation? It seem the TPC of concern depends on the stalk rather than the biochar. Perhaps, you should define TPC before the equation is presented.

7. With respect to Equations (5) and (6), if the ultimate analysis was carried out as stated in Section 2.2.2, then the elemental composition of H and O should have been captured experimentally, rather than calculated. This needs to be looked into.

8. Some explanation has been given for the change in fixed carbon. Could you elaborate on the variation in the ash content?

9. Please, check the first sentence in Section 3.3 vis-a-vis the referred table.

10. What’s the essence of the bulleted points presented in lines 300 to 312? I think this should be written properly under the required subheading or otherwise removed.

11. I will advise that you break Section 3.5 into sub-sections. This should allow proper discussion and readability of each subsection.

12. Some claims in the discussion do not seem to be supported by the methodology presented. This should be checked. For instance: "The conversion of pigeonpea stalks to biochar using this protocol resulted in a more stable form of carbon (biochar) that can withstand microbial decomposition and therefore can store atmospheric CO2 in soil." How is this shown from the methodology presented. Check for other such claims within the manuscript.

13. The results of the statistical analysis described in Section 2.3 is not presented.

Check for spelling and typing errors throughout the manuscript.

Check also for sentence construction and grammatical errors.

Author Response

AUTHOR RESPONSE TO THE REFEREE AND EDITOR COMMENTS

Thank you for your time and effort in reviewing this manuscript (MS). We have attended to all your suggestions and comments with utmost respect and acute attention.  Please note that the revised text (and figures with captions) are shown in blue ink. Please find the point-to-point response for your comments given below:

Comments to the author:

Reviewer 4:

The authors have presented some work on the production and characterization of biochar produced from pigeonpea stalk. The ideas presented are plausible but the manuscript will need extensive revision. See some of my suggestions below:

  1. References are not properly arranged. This should be thoroughly checked.

Response: Thank you very much, we have corrected in the revised MS.

  1. Be consistent with the definition of acronyms…For instance, is TPC total carbon potential or total potential carbon? Check throughout the manuscript.

Response: Yes sir, thank you for your suggestion To avoid ambiguity followed Total potential carbon (TPC) uniformly throughout the manuscript.

  1. All equations should be numbered. Some equations are not numbered.

Response: Thank you for your valuable suggesition As per reviewer suggestions checked all the equations numbers and corrected in the revised MS..

  1. Section 2.1 contains some deductions/conclusions when it simply ought to describe the methodology. Remove comments that deviate from the methodology.

Response: Thank you sir the suggestion, we have corrected these in the revised and removed the statements of conclusions from methodology section.

  1. Methodology has to be reorganized to enhance readability and correct sequence and thinking.

Response: Yes sir we agreed with you and corrected in the MS.

  1. I am concerned about the units in Equation (3), page 5. The unit on the left- and right-hand sides of the equation seem not to tally. Can you check this or probably give an explanation? It seems the TPC of concern depends on the stalk rather than the biochar. Perhaps, you should define TPC before the equation is presented.

Response: The units of total potential carbon was reverified and followed accordingly with below reference. Tesfamichael, B.; Gesesse, N.; Jabasingh, S.A. Application of rice husk and maize straw biochar for carbon sequestration and nitrous oxide emission impedement. J. Sci. Ind. Res. 2018, 77, 587–591.

  1. With respect to Equations (5) and (6), if the ultimate analysis was carried out as stated in Section 2.2.2, then the elemental composition of H and O should have been captured experimentally, rather than calculated. This needs to be looked into.

Response: Thank you for the keen observation and as per the ultimate analysis, parameters VIZ., C, N. Phosphorus (P), potassium (K), sulfur (S), boron (B), zinc (Zn), copper (Cu), iron (Fe), and manganese (Mn) were analysed. However, the instruments used (TCTN analyser, Model: PRIMACS-SNC100 and ICP-AES) were not suitable to estimate hydrogen and Oxygen. Therefore, Hydrogen and Oxygen were estimated using empirical correlation equations

  1. Some explanation has been given for the change in fixed carbon. Could you elaborate on the variation in the ash content?

Response: Yes sir I appreciate for your suggestion we have added some information “ Biochars possessing reduced ash content exhibit heightened resistance to wind-induced dispersion, rendering them optimal for transport and seamless integra-tion into soil systems [86].

  1. Please, check the first sentence in Section 3.3 vis-a-vis the referred table.

Response: Yes sir we have identifided small error their and rectified it in the revised MS, thank you sir.

  1. What’s the essence of the bulleted points presented in lines 300 to 312? I think this should be written properly under the required subheading or otherwise removed.

Response: Agreed with the suggestion we have removed the bullets in the revised MS..

  1. I will advise that you break Section 3.5 into sub-sections. This should allow proper discussion and readability of each subsection.

Response: Thanks for your suggestions, But, here we have seven fuel characters, by providing each character as sub heading leads so many sub headings. So, kindly allow us to stick to the existing format.

  1. Some claims in the discussion do not seem to be supported by the methodology presented. This should be checked. For instance: "The conversion of pigeonpea stalks to biochar using this protocol resulted in a more stable form of carbon (biochar) that can withstand microbial decomposition and therefore can store atmospheric CO2 in soil." How is this shown from the methodology presented. Check for other such claims within the manuscript.

Response: Thank you for the suggestion. A generic statement was written explaining “microbial decomposition rate of high fixed carbon content materials (Ex: biochar) is slow compared to materials with low fixed carbon (straw and crop stubbles)”. Agreeing with the reviewer’s comment, the sentence is removed as microbial decomposition study was not conducted and not mentioned as part of methodology.

  1. The results of the statistical analysis described in Section 2.3 is not presented.

Response: The data is presented as mean of triplicate samples along with standard deviation.

14. Comments on the Quality of English Language. Check for spelling and typing errors throughout the manuscript. Check also for sentence construction and grammatical errors.

Response: We have gone through the manuscript several times to fix the errors and corrected them using the premium service of Grammarly, Inc. (United States).

Round 2

Reviewer 1 Report

1. In the abstract part: Please provide explanations for all abbreviations upon their initial use, for instance, HHV (higher heating value), Fourier-transform infrared spectroscopy (FTIR), and Scanning electron microscopy (SEM).

2. Conclusion should explicitly emphasize the findings and advantages of the study. When highlighting the advantages of the BPMF method, consider using more specific figures and descriptions to enhance readers' precise understanding of its significance. For example, mention that the BPMF-produced biochar prominently exhibits a significant fixed carbon content (exceeding 80%) and notable energy density (ranging from 1.77 to 2.06 MJ/kg). In the prospect section, when discussing future possibilities, consider emphasizing the 'intriguing possibilities' of extending this method to other agricultural residues to enhance its foresight. Spotlight the potential applications of BPMF-produced biochar in soil rejuvenation, carbon emission reduction, and energy resource diversification. Concluding with the last sentence, emphasize the comprehensive contribution of the study in improving soil quality, reducing environmental impacts, and establishing a more resilient agricultural framework.

In terms of language, the article is generally clear and well-organized, but there are a few areas that could benefit from minor adjustments. For instance, there could be some refinements needed in the presentation of the conclusion section.

Author Response

AUTHOR RESPONSE TO THE REFEREE COMMENTS

Thank you for your time and effort in reviewing this manuscript (MS). We have attended to all your suggestions and comments with the utmost respect and acute attention in this second revision. We have made the necessary changes in the revised MS. We hope that referees will find the revised version more interesting and recommend a publication in Sustainability.

The revised text (and figures with captions) are shown in red ink. Please find the point-to-point response for the referee’s comments given below.

  1. In the abstract part: Please provide explanations for all abbreviations upon their initial use, for instance, HHV (higher heating value), Fourier-transform infrared spectroscopy (FTIR), and Scanning electron microscopy (SEM).

Response: Thank you for your suggestion. We have expanded these abbreviations as suggested. Earlier we have used the abbreviated form to keep the abstract within the world limit. Please find it in line no. 34 and 35 of the revised MS.

  1. Conclusion should explicitly emphasize the findings and advantages of the study. When highlighting the advantages of the BPMF method, consider using more specific figures and descriptions to enhance readers' precise understanding of its significance. For example, mention that the BPMF-produced biochar prominently exhibits a significant fixed carbon content (exceeding 80%) and notable energy density (ranging from 1.77 to 2.06 MJ/kg). In the prospect section, when discussing future possibilities, consider emphasizing the 'intriguing possibilities' of extending this method to other agricultural residues to enhance its foresight. Spotlight the potential applications of BPMF-produced biochar in soil rejuvenation, carbon emission reduction, and energy resource diversification. Concluding with the last sentence, emphasize the comprehensive contribution of the study in improving soil quality, reducing environmental impacts, and establishing a more resilient agricultural framework.

Response: Thank you for this constructive suggestion. We have modified the conclusion in the revised MS and highlighted the findings and advantages of the study. Please find it in line no. 529–540 of the revised MS.

Reviewer 3 Report

The authors have improved the manuscript significantly. As one of the important properties of biochar is the surface area and pore size, the authors should carry out the BET analysis as mentioned in the earlier comments.

Author Response

AUTHOR RESPONSE TO THE REFEREE COMMENTS

Thank you for your time and effort in reviewing this manuscript (MS). We have attended to all your suggestions and comments with the utmost respect and acute attention in this second revision. We have made the necessary changes in the revised MS. We hope that the referees will find the revised version more interesting and recommend a publication in Sustainability.

The revised text (and figures with captions) are shown in red ink. Please find the point-to-point response for the referee’s comments given below.

The authors have improved the manuscript significantly. As one of the important properties of biochar is the surface area and pore size, the authors should carry out the BET analysis as mentioned in the earlier comments.

Response: Thank you for your suggestion. As per the reviewer's earlier suggestions, we have conducted FTIR and SEM and results are included in the revised MS. We don’t have a BET analysis facility so we could not do analysis.

Reviewer 4 Report

The manuscript has been revised throughout in line with the reviewers' comments. There a few issues which the authors should note.

1. Cited references within the manuscript are still not sequential.

2. Very little grammatical errors need to be checked.

3. Some equations still not numbered. See Lines 177, 207, 213, 217, etc.

4. Where was ANOVA reported or at what stage was it applied? This is not clear from the presented results and discussion.

5. Section 3.1.4 (correct the numbering) should rather be "Elemental Composition" rather than "nutrient composition" but I do not think it should be separated from the Ultimate Analysis. If it must be separated, check the correction throughout the MS.

6. Figure 2 is not mentioned anywhere within the MS which leaves Figure 2 hanging without being referred to. Check if there are other Figures or Tables in this category.

7. Section 3.2.1 should refer to Table 3 rather than Table 4.

8. Section 3.4: The equations referred to are wrong and this should be corrected. Other references made to Tables, Equations and Figures should be checked for correctness.

9. I think the sub-heading Section 3.2.1 (Fuel Properties) should be removed.

Slight corrections required.

Author Response

AUTHOR RESPONSE TO THE REFEREE COMMENTS

Thank you for your time and effort in reviewing this manuscript (MS). We have attended to all your suggestions and comments with the utmost respect and acute attention in this second revision. We have made the necessary changes in the revised MS. We hope that the referees will find the revised version more interesting and recommend a publication in Sustainability.

The revised texts (and figures with captions) are shown in red ink. Please find the point-to-point response for the referee’s comments given below.

  1. Cited references within the manuscript are still not sequential.

Response: Thank you for your suggestion. We have now organized all the references in a sequential manner in the revised manuscript.

  1. Very little grammatical errors need to be checked.

Response:  Thank you, we have gone through the MS again and corrected the grammatical errors.

  1. Some equations still not numbered. See Lines 177, 207, 213, 217, etc.

Response: Thank you, for pointing this out. We have added the equation numbers. Please find them in lines 179, 210, 216, and 221 of the revised MS. 

  1. Where was ANOVA reported, or at what stage was it applied? This is not clear from the presented results and discussion.

Response: Thank you, we have done the paired t-test for yield, proximate, and ultimate analysis. However, as we got statistically non-significant results, we have not included it in the discussion of results. Therefore, we have removed this from the methodology section in the revised MS.

  1. Section 3.1.4 (correct the numbering) should rather be "Elemental Composition" rather than "nutrient composition" but I do not think it should be separated from the Ultimate Analysis. If it must be separated, check the correction throughout the MS.

Response: Thank you, for your valuable suggestion. We have made the necessary corrections, including renumbering 3.1.4 and changing 'nutrient composition' to 'elemental composition.

  1. Figure 2 is not mentioned anywhere within the MS which leaves Figure 2 hanging without being referred to. Check if there are other Figures or Tables in this category.

Response: Thank you, we have mentioned the Figure 2 in the line number 341 of the revised MS and also cross-checked that all the figure and table are mentioned.

  1. Section 3.2.1 should refer to Table 3 rather than Table 4.

Response: Thank you for the suggestion we have changed the tables number from 4 to 3. Please find this in line number 371 of the revised MS.

  1. Section 3.4: The equations referred to are wrong and this should be corrected. Other references made to Tables, Equations and Figures should be checked for correctness.

Response: Thank you for pointing this out, we have corrected the equations numbers and cross checked all the tables, equations and figures in the MS.

  1. I think the sub-heading Section 3.2.1 (Fuel Properties) should be removed.

Response: Thank you for your input, we have removed the sub-heading Section 3.2.1 in the revised MS.

Round 3

Reviewer 3 Report

As per my previous comments, authors should carry out the BET analysis as it is important to know the surface area, pore diameter, and pore volume for biochar. There are many facilities around India where BET can be done on a paid basis. I would suggest the authors explore these facilities and send their samples for BET analysis. 

Author Response

Response: Thank you very much for the valuable suggestion, we have tried approaching a few laboratories for BET analysis. As the private labs are charging quite a high amount besides having a high load of samples with them, we did not get confirmation from many labs. Secondly, it is a student project with limited resources available and moreover, we had already done all the suggested analyses by paying the huge costs towards those analyses. Thus, we hereby request you to kindly consider the MS without the inclusion of the BET analysis.

Reviewer 4 Report

The manuscript has been thoroughly revised. Just minute checks on grammar required.

Minute corrections required.

Author Response

Response: Thank you for the suggestion. Now, we have gone through the manuscript several times to fix the grammatical errors and corrected them using the premium service of Grammarly, Inc. (United States). Please find the corrections in the line numbers viz., 21, 33, 35, 38, 84, 128, 142, 158, 166, 174, 228, 239, 258, 260, 273, 303, 334, 361, 439, 454, 481, 497 515, 529, 530, 533 537 and 539 in the revised manuscript.

We thank all the reviewers and editor for their constructive suggestions which could help the MS to bring to the present level.